# An Integrative Review Exploring Psycho-Social Impacts and Therapeutic Interventions for Parent Caregivers of Young People Living with Duchenne’s Muscular Dystrophy

**DOI:** 10.3390/children8030212

**Published:** 2021-03-11

**Authors:** Debra Porteous, Barbara Davies, Christine English, Joanne Atkinson

**Affiliations:** Department of Nursing, Midwifery and Health, Northumbria University, Newcastle Upon Tyne NE7 7XA, UK

**Keywords:** psycho-social needs, muscular dystrophy, Duchenne’s muscular dystrophy, parents caregivers, age 13–19, family burden/care burden

## Abstract

The purpose of this integrative review was to explore psycho-social impacts and therapeutic interventions for parent caregivers of young people living with Duchenne’s Muscular Dystrophy (DMD). Electronic databases were searched for research publications between 2010 and 2020. This included Medline, CINAHL, PsycINFO, ERIC, ERC, and AMED. Four central themes emerged: Living with DMD; Knowing and telling; Transitioning; and Building resilience. The impact on parents caring for a child with DMD affected all aspects of their lives, changed over time, and had identifiable peak stress points. Unmet parental information and support needs left parents struggling in their role. Transition required changes to parenting behaviors and required adaptation and resilience. It is proposed that future investment should focus on anticipating family need, targeting intervention cognizant of predictable stress points and building resilience through social community. Parents may then be better positioned to support their child in looking forward.

## 1. Introduction

Duchenne Muscular Dystrophy (DMD) is an X-linked, degenerative, neuro-muscular disorder with an estimated male birth incidence of 1:3800 to 1:6200 [1]. The disease is characterized by a progressive degeneration of muscle fibres resulting in muscle weakness and eventual loss of ambulation. Functional dependence typically occurs in the second decade of life [2] with cardiac and respiratory complications often shortening life. Other types of muscular dystrophies, such as Becker’s muscular dystrophy (BMD) and limb-girdle muscular dystrophy (LGMD), have similar progression to DMD but a near normal life expectancy with symptoms appearing later, being less severe, and thus preserving ambulation often to mid-life.

DMD, as the most common and severe form of the disease, is the focus for this review. Disruption to daily life with DMD can commence at an early age and not only impacts the child but also the family. The nature of the disease itself and care required places a heavy burden on parent caregivers for an extended period of their lives. Improvements in supportive care for children and young people with DMD have resulted in improved quality of life and life expectancy, but, until recently, drug therapies had seen little change. This situation is rapidly changing with the emergence of new therapies that address underlying genetic defects setting a change in course for DMD treatment [1]. Antisense oligonucleotides (ASO), are new therapies that modify disease pathways by targeting underlying genetic changes. There is now potential to prevent clinical features of disease occurring with early intervention [2,3]. Another new treatment is Ataluren (Translarna); this is not an ASO, but this drug demonstrates that the non-sense mutation a genetic defect causing some types of DMD, can respond to treatment [3]. In a disease that has been seen as incurable for so long, these novel treatments offer a much-improved outlook for the future. 

## 2. Purpose and Questions 

### 2.1. Purpose of the Study 

This integrative review addressed the psycho-social needs and therapeutic interventions for caregivers of young people living with muscular dystrophy. 

### 2.2. Research Questions 

(1) What is the psycho-social impact on caregivers of young people living with muscular dystrophy?

(2) What therapeutic interventions are employed to meet the psycho-social needs of caregivers of young people living with muscular dystrophy?

(3) What factors influence accessibility of psycho-social interventions for these caregivers?

## 3. Method

The integrative review method systematically summarizes and analyzes empirical and theoretical literature to provide an in-depth understanding of a problem or phenomenon for the purpose of providing an evidence base to practice and policy development to enhance rigor in the conduct of any review, Whittemore and Knafl’s [4] modified five stage integrative review framework can provide a systematic and rigorous structure to conduct an integrative review [5,6]. This review adopted this framework and consistent with this followed the framework stages: 1. Problem identification, 2. Literature search, 3. Data evaluation, 4. Data analysis, and 5. Presentation of results [4].

### 3.1. Literature Search

Integrative reviews use more than one search strategy to enhance the quality of the review and minimize incomplete and biased results [4,5,6]. Computerized databases may only yield approximately 50% eligible studies [6]. Within this study two search strategies were made. The search was undertaken in November 2020 using a combination of the following key words: Psych-social needs, Muscular Dystrophy, Duchenne Muscular Dystrophy, Parent Caregivers, Mothers, Fathers, Family, Psycho-social needs/Interventions, Age 13–19, Family burden/Care burden. Use of Muscular Dystrophy as a key term enabled a broad search ensuring capture of all relevant papers prior to a selective search using Duchenne Muscular Dystrophy.

Search strategy 1: Electronic databases were searched for research publications between 2010 and 2020. This included Medline, CINAHL plus, PsycINFO, ERIC, ERC, and AMED. Individual databases: British Nursing Index, DARE, Cochrane Library, Joanna Briggs Institute, EThOS also completed. A manual search was conducted of the reference lists of the identified articles. To focus on the most recent publications and to identify any new emerging data the search was limited to material available between 2010–2020. Inclusion criteria identified data that addressed the aims of the study which included psycho-social needs and factors that influence accessibility of psycho-social interventions. This was achieved initially by reviewing the titles and abstracts. Any data collected that did not meet the eligibility criteria was excluded Table 1. Studies that met the inclusion criteria were reviewed and organized into a table.

*Search strategy 2:* Hand searching reference lists of retrieved articles to find relevant literature not previously identified, including grey literature. Grey literature is documentation types that includes government, academics, business, and industry in print and electronic formats that are protected by intellectual property rights.

Examples of grey literature include conference abstracts, presentations, proceedings; regulatory data; unpublished trial data; government publications; reports (such as white papers, working papers, internal documentation); dissertations/theses; patents and policies [7,8]. All data accessed was excluded from the search as in the main they were secondary sources of data or websites with resources to access.

### 3.2. Description of the Studies

The search strategy identified 979 articles: PubPsych 828, CINAHL 409, Medline 42. Additional resources identified as grey literature, 15 including government policies and Ethos. Following screening of the titles and removing duplicates (558), articles remained. Abstracts of the remaining articles were reviewed, and, if they did not adhere to the aims of the study, they were excluded. The total selected for the review was 21 (Figure 1). The literature search and the screening of the research was undertaken by the authors (D.P), (C.E), and (B.D). Verification was undertaken by (J.A). The study designs of the chosen articles were qualitative and addressed the aims of the study.

### 3.3. Quality Appraisal of the Chosen Studies

As part of any appraisal of literature, it is important to assess the methodological quality of a study. The focus is to assess the extent which a study may be biased in its design, conduct, and analysis. In this review, we applied Joanna Briggs Institute (JBI) quality appraisal checklist for qualitative research [6]. All studies chosen were subjected to rigorous appraisal and reviewed independently by the researchers. The results of this appraisal then were used to inform synthesis and interpretation of the results of this study.

### 3.4. Data Analysis

As part of the data reduction, a spreadsheet was developed. The data reduction commenced by assessing each article for its relevance to the study aims. Key themes then emerged, and then the following stage of analysis was to identify sub-themes. These sub-themes were then analyzed and synthesized. For each individual piece of data, we identified any duplications, discussed if the article met the study aims, and reached consensus of significant items.

Four key themes with additional sub themes were identified. These are Living with DMD, Knowing and telling, Transitioning, and Building resilience.

## 4. Main Findings

This integrative review posed questions around impact, intervention and accessibility. The studies analyzed were able to provide insight into the extensive psycho-social impacts on parent caregivers of young people living with muscular dystrophy. Three of the themes which emerged *Living with DMD, Knowing and telling*, and *Transitioning* represent the challenges experienced.

In contrast, there was limited evidence, within the reviewed papers, relating to the accessibility of the therapeutic interventions used to support families. Interventions were often only considered when making suggestions for future research and developments. The final theme, *Building Resilience*, reflects how the studies provided some description of interventions employed but what is not addressed is accessibility.

In the majority of studies exploring parental experiences, mothers were the main informants. It should be noted researchers mostly used the term ‘parent(s)’ to present their findings; therefore, the research reviewed did not consistently identify impacts specific to either mother or father.

What is the psycho-social impact on parent caregivers of young people living with muscular dystrophy?

The impact on parents of caring for children with DMD, changed over time, as the child’s dependency increased as they moved from childhood to adulthood. The impacts encompassed not only psycho-social aspects of their lives but also physical and financial (*Living with DMD*). In addition, communicating information about DMD emerged as a significant issue (*Knowing and telling*), along with managing transitional care (*Transitioning*).

### 4.1. Living with DMD

#### 4.1.1. Psycho-Social Impacts

Parents reported having feelings of loss, sadness and depression [9,10] as they lived with their child’s condition but identified their main psychological issue as their distress and worry for the future [11,12]. These parents experienced significantly greater stress than parents of healthy children and where their child also had difficulties in social interactions, this further increased stress levels [13]. Generally, parents felt they had coping mechanisms for day-to-day living but reported struggling, at times, with both interventions and behavioral changes [11]. In addition, parents not only worried for their affected child but also for the negative influence DMD had on the psychological well-being and social life of the siblings [10,12].

Although continually living with stress and worries for their child and family, peaks in parental stress levels were experienced at life-changing moments, notably the time of diagnosis, the point where disease progression rendered their sons immobile, when a powered wheelchair or non-invasive ventilation (NIV) became necessary, and at the death of peers [13,14,15]. These major events marked out disease progression with parents identifying their child’s loss of ambulation as one of the most difficult challenges to cope with emotionally [13]. As parents struggled with the psychological, physical, and, for some, financial impacts of their child’s disease, they often disengaged from their own hobbies and social activities [12].

Despite the many negative psychological impacts of being a parent of a child with DMD, parents repeatedly identified their experiences as positive [10,12,16,17]. Those most positive about their situation were long term caregivers and those who viewed their child as being both sensitive and talented: very few mothers reported any negative feelings about assisting their child [16]. The parents’ experiences often changed their personal life values and increased their strength and courage in facing adversity [10]. Parents were themselves ageing as their child’s dependence was increasing, but they felt increased confidence from having raised their children and satisfaction with their work as caregivers [17]. In contrast, some parents who had developed health problems themselves (sight issues, back problems and hypertension) had concerns about not being as able to care for their child as previously [17].

#### 4.1.2. Physical and Financial Impacts

Home management of their child’s care was a demanding role for parents and this physical care burden was perceived as greatest where the child had suffered with the disease longer, had lower functional ability, and was more dependent on caregivers [10,12]. The demands on parents were extensive with many reporting night-time wakening to give care and respond to equipment alarms, especially in the later disease stages due to the child’s immobility and need for non-invasive ventilation (NIV). Malfunction or dislodgement of NIV can be fatal for the child should the care giver not intervene, and these nighttime care demands negatively impacted the quantity and quality of sleep for parental caregivers, with those less experienced being most adversely affected [15]. Perception of the extent of the care burden was linked with parental access to social contact and support from friends, family, and professionals, especially in emergency situations [10,12].

The nature of the disease meant there was a growing physical dependency on parents as the child matured and an increasing financial burden for some families [15,18]. As their child grew into a young man, parents anxiously anticipated their own ageing, retirement, and the changes in family relationships and structures as their other children grew up and had families [17]. For parents, these events coincided with increasing care needs and financial burden for them alongside the loss of their own primary caregiver role [17].

Parents admitted harboring some regrets for the life constraints that DMD had imposed on them, some of which were financial [17]. Some studies have demonstrated that families with a child with DMD have a lower than national median income with many costs associated with care provision resulting in substantial economic burden for families [18,19], and, in low socio-economic countries, the impact on families of the disease was even greater [14]. Economic worries were real, and families found it difficult to escape poverty or even think about how to increase their income [17].

### 4.2. Knowing and Telling

This theme recognizes the significance of the moment when parents were first told of their child’s diagnosis. It was at this stage that parents were told of their child’s prognosis with this genetic, degenerative disease and mothers first found out of their own potential carrier risk and that of their daughters [20].

Mothers felt responsible for telling their daughters about their carrier risk and imparted critical pieces of knowledge to them at key developmental stages, for example, starting high school, turning sixteen, or being in a serious relationship [20]. This lengthy, complex process allowed daughters to assimilate the knowledge as they matured. Six levels of disclosure of information were identified: condition; genetics; carrier risk; carrier test requested; reproductive options and carrier testing; life expectancy. Mothers who told their daughters the most information often did this through unplanned conversations as they responded to questioning. Lack of knowledge about advanced genetic reproduction options and beliefs about timing of discussions acted as key communication barriers for mothers. Similarly, reduced life expectancy was considered too difficult a topic to talk about so was generally avoided [20].

It was proposed that, following diagnosis, parents would benefit from having written information about DMD which should include key facts, a summary of reproductive options and advice about how to tell daughters this critical information. In addition, genetic counseling and psychological support for both daughters and mothers may help to mitigate the long-lasting guilt and blame often felt by carriers [14,20].

Discussion of potential carrier status with daughters posed specific challenges, but, equally, parents struggled about talking to their sons about their disease and its progression [14]. Parents reported that their own lack of knowledge and understanding about DMD, its progression, advances in treatment, and access to supportive services often prevented them effectively communicating with their child [14]. One parent had told his son he had ‘weak muscles’ when his child was upset, and he could no longer run or play, while other parents avoided conversations about disease progression altogether [14]. Parents wanted more knowledge, so they were better equipped for these conversations [14], and, even at transition to adult services, parents still highlighted their need to understand DMD and be up to date, in readiness to be a ‘back-up’ carer [21]. These findings demonstrate that, throughout their journey, parents have information and support needs that are unmet, leaving them struggling to communicate and help their children who are affected by DMD.

### 4.3. Transitioning

Increased life expectancy in young people with DMD, due to interventions, such as ventilation, is allowing caregivers to be aspirational about their child’s future. Transition has been described as not only moving from pediatric to adult care, from childhood to adulthood, but also in terms of disease progression. During teenage years, young people begin to plan their futures, including continuing education, entering the work of work, and expanding their social relationships. For boys with muscular dystrophy, trying to emerge into adulthood occurs at a time when their physical dependence on others is increasing.

The age of transition occurs much later than the normal population [21]. The experience of young people with disabilities is one where they are falling behind their peers in fulfilling adult social roles due to reduced opportunities, lack of expectation, and overprotectiveness from those involved in their care. For successful family functioning, during transition, balance needs to be achieved between the needs of the young person and those of the rest of the family [22].

Parents themselves needed to change their behaviors during transition, moving from a ‘manager’ role to one of ‘consultant’ as their son matured [23,24] but remaining a strong influence [21,25]. Having been a primary caregiver for a prolonged period brings with it additional challenges [17]. Entrusting their son’s care to aides was difficult as parents fundamentally believed that no one except them could always put him first. Parents became deeply concerned about their son’s future as they became older themselves [21]. At this time of transition, the majority were dependent on mechanical ventilation, and parents lived with a sense of impending crisis knowing their son’s life was reliant on the mechanical ventilator and care. Parents acknowledged that, whilst trying not to meddle, they knew they swayed their son’s decisions because of their own anxieties.

It is possible that, as DMD progresses, parents may feel overwhelmed by their caring role and too exhausted to be involved in social activities. This situation may lead to a vicious cycle of events where parents gradually reduce their social engagement to cope, but, in fact, this may expose them to greater burden as time progresses, with further social withdrawal impacting on transition [12,20].

The role of parents in transition, is multifaceted. Whilst maintaining their role as a “lifeline”, they need to adapt by changing parenting and caring behaviors [21]. There is a requirement to increase the flexibility of family boundaries to permit development of children’s independence. Parents need to develop new adult relationships with their children [22].

What therapeutic interventions are employed to meet the psycho-social needs of parent caregivers of young people living with muscular dystrophy?

A limited number of the reviewed papers outlined potential therapeutic interventions aimed at addressing psycho-social needs of parent caregivers. The key theme, *Building resilience*, underpins interventions identified in the literature which focused on adapting, well-being, socializing, and escaping.

### 4.4. Building Resilience

#### 4.4.1. Adapting

Resilience is not just the ability to bounce back from adversity but the process of adapting [26]. There is a need to foster the mother’s resilience using psycho-social interventions aimed at improving acceptance by identifying the positive aspects of living with DMD rather than just the burden and deficit [10,27,28].

Psycho-social support should start when the child is young, and those involved in caring for families should assess unmet needs. Being proactive in identifying the need for help and understanding the family’s fears and uncertainty also helps in the identification of resources needed to prioritize and customize interventions building on family strengths [29].

#### 4.4.2. Wellbeing

Parental health is seen as a necessity for family adaptation [10,28,30]; however, the requirements for good parental health are varied. The family environment can contribute to or mitigate burden [17,29]. Family and partner support are seen as important [29]. It has been found that an intact family structure may influence family hardiness and the provision of emotional support [29,30,31]. If the parent is single, or the child not of school age, there is greater need for social and professional support to manage care and minimize burden [10].

Parents of children with DMD experience higher levels of stress than other parents [28,32], and it is reported that distressed or depressed parents may become frustrated and see their children as more of a burden [28]. Where stress is related to their child, or problem behaviors, such as clinginess and poor socialization skills, interactions between mother and child also become more stressful [32]. This correlation between parental stress and psycho-social adjustment can lead to a decline in good parenting skills and poorer coping mechanisms [28].

Good parental health and management of stress is reliant on responsive community services [9,10,30] and supportive health professionals [29]. Attendance at support groups, where parents can discuss their fears and anxieties, gain advice on resources, and expand their knowledge, is seen to reduce levels of stress and aid good psycho-social adjustment [28].

#### 4.4.3. Socializing

Creating opportunities for socialization is an important intervention, not just for the young person but the parent [9,23,24]. Social support is frequently used by caregivers as a coping strategy [24], and, whilst it can be challenging, the benefits are clear [23]. Parents found that engaging in support groups gave them access to practical advice, emotional support, and helped them to understand their child’s condition better [24]. Sharing and improving knowledge is linked with active coping [6], giving parents power and the voice to advocate for their child [9].

Access to social support and the opportunity to socialize outside of the home can allow the caregiver to escape (24) or get away (8) from caregiver responsibilities. This becomes more important as DMD progresses as there is a potential for caregivers to reduce social engagement to cope with caring responsibilities and, thus, further increase burden. To avoid this vicious cycle of events, there is a need to build upon existing social supports to improve the psychosocial outcomes for families coping with the effects of DMD.

#### 4.4.4. Escaping

There was acknowledgement that parents could benefit from having some time-out from their role and responsibilities. Escaping the ongoing daily pressures could provide some well needed relief for parents. So, paid employment may not only have financial benefits but also could be a means of escaping care burden even if only for short periods of time [10,30]. This time out is important.

Parents also identify exercise and self-care activities as being necessary for their overall health [29]; however, many neglect hobbies [10] and spend little time on social activities and rest [31]. Over time, as their child matured and became more dependent, the parents’ neglect of their own well-being often increased [12], and their quality of life deteriorated [33]. Respite care can improve caregiver burden [30], but uptake is low [28].

What factors influence accessibility of psycho-social interventions for these caregivers?

Accessibility of psycho-social interventions for caregivers was not a central focus of the studies; therefore, this research question was not fully addressed, indicating a need for further research in this area. Researchers noted variation in the level of professional and social support for parents within their studies, but factors influencing this situation were not explored. For example, it was noted that uptake of respite care is low [28], but the reasons for this are not clearly established. It is, however, suggested that it could be ‘too time consuming to organize’ or due to maternal anxiety about relinquishing care [29]. It is possible there are many issues underlying the decision making in accessing respite services; therefore, to fully understand this, future research needs to concentrate on uncovering the potentially complex influencing factors.

Similarly, there is recognition that good parental health and management of stress is reliant on responsive community services [9,10,30] and supportive health professionals [29]. However, some families struggled to access resources and had to fight for services, identifying a lack of joined up thinking as a barrier to consistent care [9]. Without a fuller understanding of these critical access issues, it is difficult to draw any conclusions regarding factors influencing this situation.

## 5. Discussion

The review findings demonstrated the impact on parents caring for children with DMD changed over time and encompassed psycho-social, physical, and financial aspects of their lives. Parents lived with continuing levels of stress and worry which peaked at diagnosis and with each milestone in their child’s disease progression. These key markers of life-changing moments can be predicted and include their child’s loss of mobility, need for enteral feeding, need for NIV, transition to adulthood, and when their friends die. This heavy care burden for parents and impact on parents’ lives has previously been recognized in the literature [22]. However, this review extends our understanding of the pressures on parents, particularly at identifiable, predictable life-changing moments, and this knowledge is critical to development of improved parental support and information provision.

What is evidenced within the review is that some families have more help and support around them. Social circumstances, availability of professional and family support and information about DMD influenced family experiences of living with the condition. Several studies reported unmet parental needs changing throughout the child’s disease trajectory, impacting upon communication, education, and coping. The parental dilemmas in communicating with their child about disease progression and prognosis, uncovered in the review, echoes previous findings about disclosures to children with a cancer diagnosis [34,35,36,37]. Adoption of suggested communication improvements from cancer care research could be equally beneficial for parents of children with DMD, as the role of information gatekeeper seems common to both parent groups.

Gibson et al. [37] found that there was often a mismatch between professionals’ communication and parental information needs within individual changing circumstances. So, it may be useful within the context of DMD for professionals to aim to better match their communications with parental needs, as they too experience altering circumstances as their child’s disease progresses.

Despite the many demands on parents some were still able to find positives in their experiences. They articulated satisfaction in their caregiving having raised their child and development of confidence, personal strength, and courage in adversity were reported. No mothers identified negative feelings in caring for their child. These positive evaluations of parenthood, whilst they managed their heavy care burden, mirrors existing research findings from parents of adult sons with DMD [38]. Lazarus and Folkman’s Transactional Model of Stress [39] may offer explanation for these findings, in that parents with adequate resources to meet the demands placed on them may then see positives in their experiences.

Findings from the review suggested that transition into adulthood brings additional challenges for the young person and their parents. The timing of transition occurs later for the young person with DMD and requires both parties to adopt different behaviors to facilitate role change. The parents need to ‘let go’, and, by doing so, the young person can learn to direct their own health care. Transition into adulthood brings additional challenges for the young person and their parents. One important aspect of this, suggested by Sonneveld [40], is recognizing that the changes in responsibility are managed gradually. As parents begin the process of ‘letting go’, they will feel a loss of control [26,41]; however, they can be reassured that, to manage their transition successfully, the young person will need to use their parent as a ‘safety net’. They are not abandoning their child but allowing them to grow using their new role as facilitator.

It is interesting to note that analysis of the reviewed papers demonstrated a lack of reference to palliative care despite DMD being a progressive disorder leading to reduced life expectancy. This could be because parents view a changing approach to care as a natural trajectory and may not necessarily understand terminology. Terminal care, palliative care, supportive care, hospice care, and end of life care are all terms that may become confused, feared, and misunderstood [42] or avoided. The lack of consensus and understanding of the constantly changing nomenclature can undermine care provision and confuse the way care is delivered to people who are most vulnerable at the end of their lives [43,44]. This was not a focus of the review but is worthy of further exploration as to how the subject is approached with parents of children with DMD.

## 6. Conclusions

This integrative review highlights areas for improvement in meeting the parent’s psycho-social needs, critical to improving parents’ experiences is anticipating their needs in advance. The review has identified major stress points where parents need additional support and information to help them to adapt during challenging periods. Armed with this knowledge, health care professionals, utilizing a needs assessment approach, can target interventions in a timely manner to promote adaptation and build resilience. It is proposed that, by more effectively supporting the parent(s), this will potentially have positive effects on family functioning.

Medical advances have changed the landscape of care and have allowed parents to become more aspirational in planning for the future. The advent of revolutionary new drug therapies should bring dramatic impacts for families. This was not an aspect covered in this review as the children had not yet benefitted from these new therapies.

What is needed is more focus on helping parents adapt to their growing child and change their role back from caregiver to parent, allowing the young people to also be aspirational about their future.

Evidence suggests that there is benefit in challenging perspectives so that the focus is not always on deficit and burden but on positivity and opportunities. The nature of the disease has the potential to isolate and reduce social skill development essential for transition to adulthood. As interactive situations with peers and adult role models with DMD is seen to have positive impacts on both young people and parents.

It is proposed that future investment should focus on anticipating family need, targeting intervention cognisant of predictable stress points, and building resilience through social community. Parents may then be better positioned to support their child in looking forwards.

## Figures and Tables

**Figure 1 children-08-00212-f001:**
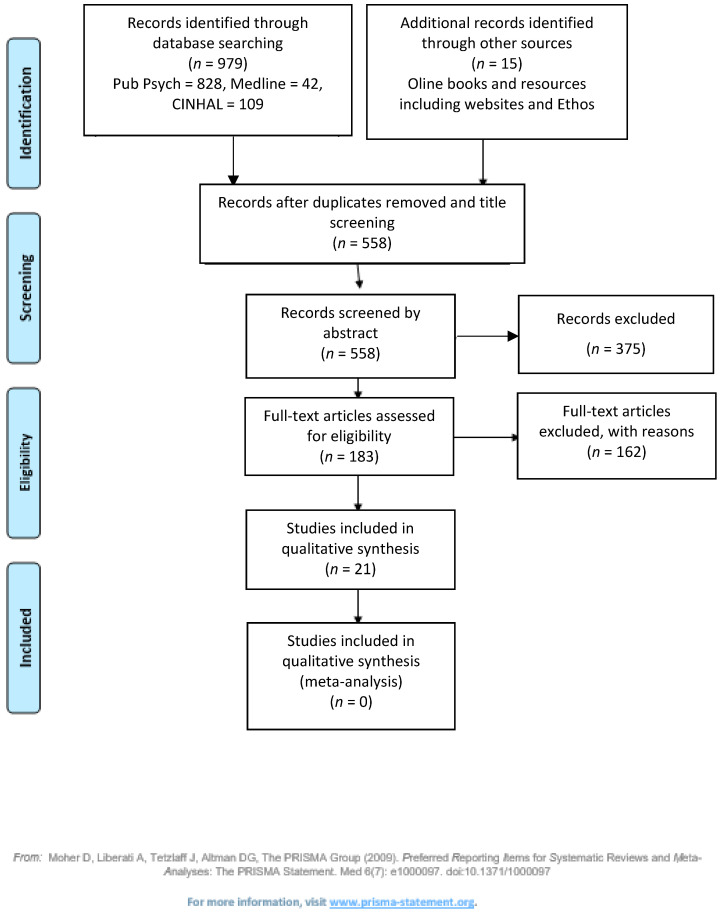
Search Strategy.

**Table 1 children-08-00212-t001:** Inclusion and exclusion criteria.

Inclusion Criteria	Exclusion Criteria
Primary sources 2010–2020	Paper in press
Psycho-social needs Factors that influence accessibility of psycho-social interventions Original publication in English Meets aims of study	Abstracts, conference proceedings Websites with resources available Secondary source or meta literature

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
