# Peer review of "An Integrative Review Exploring Psycho-Social Impacts and Therapeutic Interventions for Parent Caregivers of Young People Living with Duchenne’s Muscular Dystrophy"

_children, 2021, doi:10.3390/children8030212_

Round 1
Reviewer 1 Report
The review describes about the psycho-social impact of parents of the DMD patients. The review is well written and the method is sound in terms of keywords used for search strategy. However, it needs significant improvement in describing the main findings. I have some suggestions as descibed below
- The authors have already defined the research questions but the main findings are not presented accordingly. It can be presented in a better way.
- It will be helpful if the author will provide tables describing about the effect on father and mother separately.
- Thearpeutic interventions and accessibility of the psycho-social intervention should be described separately in the main findings.
- In the introduction, the author should give a brief introduction about DMD, BMD and Limb-girdle MD.
- The effect of DMD is different on parents as it progresses. The author should also describes about its effect on each parent. For example, what is the effect of DMD during childhood, adolescent and adulthood?
Reviewer 2 Report
Line 20: The term “life-limiting” for DMD is inappropriate.
Line 54: Since this paper focus on discussing Duchenne Muscular Dystrophy, I am wondering why authors used the keywords of muscular dystrophy, a broader scope of the selected field.
Main criticism: As indicated in the paper, the selection ranges from 2010 to 2020; however, during the past decade, the care method of muscular dystrophy, especially DMD, has been improved greatly. However, the authors just focused on the discussion of respiratory support (NIV) and nutritional issues. Besides, some novel therapies, such as ASO or gene therapy, are undergoing clinical trials. Of note, one approved drug, Translarna (Ataluren), has been applied to non-sense mutation DMD patients. However, the authors did not discuss the impact of these novel therapies on DMD. I suggest that the authors include several references regarding DMD therapies and discuss their psycho-social effects more specifically.
The authors should discuss the influencing factors of selecting palliative care, when and who makes the decision.
Round 2
Reviewer 1 Report
The authors have addressed all my concerns and the quality of manuscript has improved significantly.